# Clay Tailings Flocculated in Seawater and Industrial Water: Analysis of Aggregates, Sedimentation, and Supernatant Quality

**DOI:** 10.3390/polym16101441

**Published:** 2024-05-20

**Authors:** Williams H. Leiva, Norman Toro, Pedro Robles, Gonzalo R. Quezada, Iván Salazar, Ricardo Jeldres

**Affiliations:** 1Facultad de Ingeniería, Arquitectura y Diseño, Universidad San Sebastián, Concepción 4030000, Chile; williams.leiva@uss.cl; 2Faculty of Engineering and Architecture, Universidad Arturo Prat, Iquique 1100000, Chile; notoro@unap.cl; 3Escuela de Ingeniería Química, Pontificia Universidad Católica de Valparaíso, Valparaíso 2340000, Chile; 4Escuela de Ingeniería Civil Química, Universidad del Bío-Bío, Concepción 4030000, Chile; grquezada@ubiobio.cl; 5Departamento de Ingeniería Civil, Universidad Católica del Norte, Antofagasta 1270709, Chile; isalazar@ucn.cl; 6Departamento de Ingeniería Química y Procesos de Minerales, Facultad de Ingeniería, Universidad de Antofagasta, Av. Angamos 601, Antofagasta 1240000, Chile; ricardo.jeldres@uantof.cl

**Keywords:** clay, seawater, industrial water, sedimentation, rheology

## Abstract

High-molecular-weight anionic polyacrylamide was used to analyze the effect of kaolin on the structure of particle aggregates formed in freshwater and seawater. Batch flocculation experiments were performed to determine the size of the flocculated aggregates over time by using focused beam reflectance measurements. Sedimentation tests were performed to analyze the settling rate of the solid–liquid interface and the turbidity of the supernatant. Subsequently, a model that relates the hindered settling rate to the aggregate size was used to determine the mass fractal dimension ($D_{f}$). Flocculation kinetics revealed that greater amounts of kaolin generated larger aggregates because of its lamellar morphology. The maximum size was between 10 and 20 s of flocculation under all conditions. However, the presence of kaolin reduced the settling rate. The fractal dimension decreased with the increase in the kaolin content, resulting in the formation of irregular and porous aggregates. By contrast, factors such as the flocculation time, water quality, and quartz size had limited influences on the fractal dimension. Seawater produced a clearer supernatant because of its higher ionic strength and precoagulation of particles. Notably, the harmful effect of clays in seawater was reduced.

## 1. Introduction

Water use must be optimized because of its scarcity, especially in arid industrial areas. Solid–liquid separation is performed in the mineral extraction processes to promote dewatering from the tailings produced in flotation and to introduce it back into the upstream processes, which reduces freshwater consumption. This separation is performed using various thickener technologies [1] to increase the concentration of solids in tailings through accelerated particle sedimentation using chemical reagents of various types.

Various induced agglomeration mechanisms, such as charge neutralization, patch–charge interaction, and polymer bridges, have been devised [2,3,4,5]. A polymer bridge, which consists of promoting particle agglomeration using high-molecular-weight polymers, is widely used in dewatering mine tailings. These reagents are adsorbed to the surface of several particles simultaneously through hydrogen bonds and cationic bridges [6,7], generating larger structures that settle quickly, forming different phases in a thickener: (1) clarified water zone, (2) sedimentation zone, and (3) compaction zone.

An operational challenge is the presence of clays in copper and copper–gold minerals [8], which lowers settling rates and increases the values of rheological parameters. Thus, the draining and transport of thickened tailings become difficult [9,10]. With kaolinite being the most representative mineral, clays of the kaolin group are commonly found in mineral deposits and exhibit approximately hexagonal, fine plate-like hydrated aluminum silicate whose face length is 10 times its edge thickness. In an aqueous suspension, kaolinite particles exhibit a negative charge on basal surfaces. Under acidic conditions (pH 6), the alumina exposed at the edges of the plates binds to hydrogen ions and assumes a positive charge [11], which causes electrostatic attraction between edges and faces, producing aggregates as “houses of cards” that are highly expanded. Under alkaline conditions, the edges become neutral or negatively charged, and particles deflocculate as long as the electrolyte concentration in the solution is low. At high electrolyte concentrations (at high and low pH), electrostatic repulsion (or attraction) between particles reduces because of double-layer compression or ionic shielding from surface charges. Under these conditions, residual valence forces on the particle surface cause the particles to adhere to each other along their basal surfaces, forming a “pack of cards” structure [12,13].

The inclusion of pH-modifying reagents and salts considerably affects the sedimentation behavior of clay particles in tailings. This phenomenon directly affects the turbidity of the water subjected to recycling processes. Nasser and James [14] observed that kaolinite particles exhibit a flocculated conformation in acidic environments and in the absence of polymeric flocculant because of the intense electrostatic interactions between the opposite charges of the basal planes and edges. By contrast, under alkaline conditions, the particles exhibit a dispersed conformation at a NaCl electrolyte concentration of 0.001 M, whereas they exhibit a flocculated arrangement at NaCl electrolyte concentrations of 0.1 and 1 M. Jeldres et al. [15] identified a critical NaCl concentration at which kaolinite flocculation reached its maximum efficiency, achieving the highest initial settling rate. This phenomenon could be attributed to the formation of a floc network because of the competition between two mechanisms despite salinity variations: (i) charge detection of anionic particles and flocculants by salt counterions and (ii) shielding of the active sites in the flocculant by salt cations, inducing the folding of the polymer into compact structures with a rounded conformation. The first mechanism promotes flocculation, whereas the second suppresses flocculation.

The structure of flocculated clay mineral aggregates is critical for the dewatering of mine tailings. Although an increase in the floc size can increase the settling rate, the compact consolidation of the resulting sediment can decrease if the flocs are not sufficiently dense [16]. For example, aggregates can have the same size; however, if their structure differs because of the different arrangements of particles during their formation, the settling rates differ considerably [17]. To minimize the water retained within an aggregate, primary particles should bind in the initial stages of thickening to form large, high-density aggregates so that they settle faster with a lower water content within the aggregate. Yoshihiro Kuroda et al. [18] obtained very high porosity kaolinite particle flocs (98%) with an almost spherical shape, which were explained by a face–edge association and coagulation of angular parts in a kaolinite suspension at 0.1 M NaCl solution and pH 10.

The structure of the aggregates can be represented using the fractal dimension [19], and its influence on flocculation has been demonstrated [20,21]. Factors such as mixing [22], pH, and coagulant dosage [23,24] can affect the fractal structure. This parameter influences aggregate permeability and density, which are related to the collision rate, settling rate, and floc strength. A destabilized suspension was subjected to the optimal hydrodynamic force and the cohesive force. These optimal values depend on the shear rate and stirring time for the suspension and the interaction forces (attractive or repulsive) given by the composition and concentration of particles and the addition of reactants for the second force. During destabilization, the suspension undergoes aggregation, breakdown, and, in some cases, restructuring [25,26]. During aggregation, the chemical conditions of the suspension remain constant, resulting in the cohesive force not changing during the process, allowing the hydrodynamic force to dominate the aggregation, rupture, and restructuring processes. Studies have analyzed the structure of aggregates in suspension by quantifying the fractal dimension under various hydrodynamic forces. These studies resulted in aggregates becoming dense, more compact, and less porous with an increase in the shear rate [27,28,29].

Studies have focused on the relationship between the structure of aggregates and hydrodynamic conditions to evaluate water treatment industries for human consumption, ceramics, or biological processes [30,31,32,33]. However, these studies are generally conducted under dilute conditions with a single solid component whose composition does not vary. However, in the mining industry, tailings typically have high concentrations of solids (Cp > 12%) in addition to variations in their mineralogical composition that lead to different behaviors in the dewatering process. The objective of this study is to relate the effect of the kaolinite concentration on the restructuring of aggregates flocculated with an anionic polyacrylamide, subjected to various salinity conditions, flocculant doses, and reaction times between the tailings–flocculant mixture. The mixing time is related to the flocculation kinetics and fractal dimension of the aggregates using focused beam reflectance measurement (FBRM) techniques. The results of this study can benefit mining industries that wish to use seawater directly or after pretreatment in their extraction processes.

## 2. Materials and Methods

### 2.1. Materials

Bi-compositional synthetic tailings in quartz (Qz) and kaolin (Kao) solids were prepared at Qz:Kao ratios of 9:1, 8:2, and 7:3. Quartz particles were supplied by Donde Capo (Chilean store) with a density of 2.65 g/cm^3^ and were then sieved until the following two size distributions were obtained: (i) fine and (ii) coarse (Figure 1). Kaolin particles were purchased from Ward’s Science, and their composition was determined using a Bruker X-ray diffractometer, the D8 Advance, and analyzed using the Powder Diffraction File of the International Center for Diffraction Data software. The spectrum (Figure 2) indicates a majority of kaolinite above 10% and a minority of quartz. Particle size distributions (volumetric distributions) were determined through laser diffraction (PSA 1190, Anton Paar) for both minerals, which indicated that kaolin particles were smaller than both quartz distributions (Figure 2). The dispersing phase of the synthetic tailings consisted of two types of water: (i) process water (low salinity) and (ii) seawater (high salinity), both artificial. The composition of the process water was prepared at 0.01 M NaCl and 0.005 M CaCl_2_ in distilled water, whereas the composition of the seawater prepared according to Mobin and Shabnam is presented in Table 1 [34]. Ultra-high pure chemical reagents obtained from Merck, Darmstadt, Germany, were added to distilled water.

The flocculation process was performed in an MG15 cylindrical glass with dimensions of 10.8 cm in height, 9.3 cm in diameter, a conical base 1.4 cm in height, and a discharge hole 1.2 cm in diameter in its center. The pH of all prepared synthetic tailings was adjusted to pH 7.5 ± 0.2 with NaOH. A commercial high-molecular-weight acrylamide/acrylate copolymer SNF604 (>18 × 10^6^ g/mol) was used as a flocculant.

### 2.2. Flocculation Kinetics

Synthetic tailings pulps were prepared in the MG15 cylindrical vessel with a mass of 250 g and 10% weight in solids, where 225 g of fluid phase corresponded to the type of water in the flocculant solution. First, a 220 g prepulp consisting of all the solids (25 g) and 195 g of water, as appropriate, was prepared under turbulent stirring for 10 min to ensure homogeneity before FBRM probe introduction and pH adjustment. Next, the mixing intensity was adjusted, and the flocculation kinetics were measured. After measurement for 30 s, the flocculant solution according to the corresponding dose and the remainder of the corresponding type of water were added simultaneously until 250 g of pulp was obtained. Each test lasted 3 min. This methodology was performed to maintain the same concentration of solids for all pulps at the time of the addition of the flocculant.

FBRM technology was used to track the size distribution of primary particles and flocculated aggregates. FBRM is a novel method in which a sensor measures the reflection pulses of a laser beam on suspended particles and makes particle size measurements from 0.25 to 1000 μm. In this technique, data are acquired online and in real time to provide particle size data and suspended particle population trends. This technique does not require sampling or isolation, which contributes to changes in particle size and distribution because of breakage or agglomeration. However, this technology provides a chord length distribution rather than an aggregate diameter distribution. The laser beam meets the solid particle in the focal plane, which subsequently generates backscattered light. This time is proportional to the path that the laser beam travels over the particle, whereas the path corresponds to the straight-line segment that joins the edges of both sides of the solid particle, known as a “chord.” Therefore, the length of the chord of the chode can be calculated by multiplying the time measured between the emission of the laser beam and the reflection with the speed of movement of the laser. The iC FBRM software, which allows processing the data through histograms that record the chord lengths in channels of selected size ranges every 2 s and easily captures the temporal changes of particulate systems, was used for detailed analysis of the particles and aggregates.

### 2.3. Sedimentation

After the flocculation process was completed, the pulp was transferred from the bottom of the MG15 cylindrical glass to a cylindrical tube 24.7 cm in height, 3.5 cm in diameter, and 400 cm^3^ in volume. When the complete pulp was unloaded, the pulp was immediately homogenized to begin measuring the settling rate using a video camera and subsequent analysis.

### 2.4. Fractal Dimension

Heath et al. [35] used the following equation to calculate the mass fractal dimension of the aggregates
(1)$$U_{h}=\frac{\overset{¯}{d_{agg}^{2}}g(\rho_{s}-\rho_{l}){\left(\frac{\overset{¯}{d_{agg}}}{\overset{¯}{d_{p}}}\right)}^{D_{f}-3}}{18\mu }{\left(1-\emptyset_{s}{\left(\frac{\overset{¯}{d_{agg}}}{\overset{¯}{d_{p}}}\right)}^{3-D_{f}}\right)}^{4.65}$$
where:
$U_{h}$: hindered settling rate, m s^−1^.$d_{agg}$: aggregate size (floc), m.$d_{p}$: size of the primary particle (before addition of flocculant), m.$\rho_{s}{\;y\;\rho }_{l}$: densities of the solid and liquid phases, respectively, kg m^−3^.g: gravity acceleration (9.81 m s^−2^).μ: fluid viscosity, N s m^−2.^.$\emptyset_{s}$: volumetric concentration of the solids.$D_{f}$: mass length fractal dimension.

The equation reveals four variable terms: (i) the hindered settling rate determined experimentally by video camera analysis, (ii) the size of the flocculated aggregates determined by the squared-weighted mean chord length given by the FBRM tool, (iii) the density of the solid phase, and (iv) the size of the primary particle. The density of the solid phase and its size depend on the kaolin content. Parameters such as gravity acceleration, liquid-phase viscosity, and volumetric concentration of solids are known and remain constant independent of the hydrodynamic conditions and kaolin content that the pulp is subjected to (Table 2). When all terms are known, the fractal dimension of the mass length is determined by adjusting Equation (1) to the experimental results using the least squares method.

## 3. Results and Discussion

### 3.1. Clay Tailing Flocculation Kinetics

Flocculation kinetics experiments reveal the variation in the size of particle aggregates formed by flocculation after the addition of the SNF 604 reagent. The size of the aggregates, regardless of whether flocculated or deflocculated, is expressed by the chord length weighted mean squared according to the following equation:(2)$$Mean\;square-weighted\;chord\;lenght=\frac{\sum_{i=1}^{k}n_{i}M_{i}^{3}}{\sum_{i=1}^{k}n_{i}M_{i}^{2}}$$
where $n_{i}$ is the particle count in the chord channel, and $M_{i}$ is the channel midpoint.

Studies have confirmed that the use of FBRM technology is a valuable tool for understanding flocculation phenomena in mine tailings [36,37,38,39,40,41]. This phenomenon allows determining: (i) the optimal dose of flocculant, (ii) the size distribution of the flocculated aggregates over time, (iii) population balance studies, (iv) estimation of the resistance of aggregates in shear conditions, and (v) research on floc restructuring.

Figure 3A–D represents the flocculation kinetics of three types of tailings with various quartz—kaolin ratios (Qz:Kao = 9:1, Qz:Kao = 8:2, and Qz:Kao = 7:3) in the presence of fine quartz in process water, fine quartz in seawater, coarse quartz in process water, and coarse quartz in seawater, respectively. All these tailings were subjected to flocculation at pH 7.5 and stirred at 180 rpm using 25 g/t of SNF604.

In all cases, the same trend was observed: as the kaolin content increased, the size of the flocs also increased. Kaolinite, the primary component of kaolin, has a lamellar structure consisting of a permanently negatively charged silica basal face surface over a wide pH range and a positively charged alumina edge surface at neutral and acidic pHs [42,43,44,45]. This phenomenon implies that, depending on the chemical conditions of the suspension, kaolinite particles can associate in three ways: (i) edge–edge (E–E), (ii) edge–face (E–F), and (iii) face–face (F–F) [46].

In the crystalline structure of quartz, each silicon atom is tetrahedrally coordinated by four oxygen atoms. These [SiO_4_] tetrahedra do not carry a net negative charge within the complete context of the quartz structure, as the oxygen atoms are shared between adjacent tetrahedra, resulting in a globally neutral chemical composition of SiO_2_ [47]. This phenomenon suggests that quartz assemblages form more compact aggregate structures compared with any kaolinite assemblages, generating aggregates of smaller volume (and therefore size) when the kaolin content is lower. However, Figure 4D shows that the floc size for tailings Qz:Kao = 7:3 is smaller than the flocs for tailings Qz:Kao = 8:2 after reaching the optimal size for both cases. This phenomenon, which differs from the other behaviors, can be explained by two reasons: (i) larger quartz produces less probability of adsorption on kaolin particles, generating weaker quartz-kaolin aggregates that are easier to break as a result of the hydrodynamic forces generated by the mixing intensity and (ii) seawater favors cationic bridges between particles, which in the case of kaolin-kaolin particles implies a structure in the form of a house of cards that, when agglomerated by polymeric bridges, generates more porous structures that are easier to break.

The figures provide the following additional information: (i) the maximum floc size was larger in process water (low ionic strength) than in seawater (high ionic strength), and (ii) it was larger for coarse quartz than for fine quartz. In the first case, salinity influences floc conformation. Jeldres et al. [15] determined an optimal salt concentration to achieve the best flocculation. They studied the flocculation process of kaolinite suspensions at natural pH (between 5.8 and 6.0) and a salinity range between 0 and 0.5 M NaCl using a high-molecular-weight anionic acrylamide copolymer as a flocculant. In this case, the best flocculation, expressed by the highest sedimentation rate, was achieved at a salinity of only 0.001 M NaCl (low ionic strength), whereas higher salinity resulted in lower sedimentation rates. The presence of saline agents produces two interaction mechanisms with flocculated suspensions: (i) a cationic bridge between the anionic particles and the anionic active sites of the flocculant, promoting flocculation and (ii) shielding of the active sites of the flocculant, producing flocculant curling and damaging flocculation. These results are consistent with those of other studies where low salinity revealed superior flocculation performance compared with higher saline environments, such as seawater, for clay and tailings systems [48,49,50]. For the second case, a larger particle (coarse) exhibits a greater probability of colliding with the flocculant than a small particle (fine).

Hogg [51] developed a model that predicted that for particles that are large relative to the adsorbed polymer molecules, collision efficiencies are very high, close to 100%, over a wide range of surface coverages (0.1 < θ < 0.9, typically). Generally, collision efficiencies increase with an increase in particle size and decrease with an increase in the polymer molecular weight. Hogg [52] analyzed the probability of a polymer molecule being adsorbed per particle using a high-molecular-weight flocculant and fixed doses on particles of different sizes. For medium-sized particles of 0.5 µm, this probability was considerably less than 0.1, whereas for 1 µm particles, it was approximately 0.2.

Experimental studies have supported a higher flocculation efficiency for larger particle sizes. For example, Grabsch et al. [53] compared the effect of the particle size on the dosage response for seven pulps at a concentration of 150 kg m^−3^. They observed lower flocculant demand and faster sedimentation as the particle size became coarser, especially in a high-grade iron ore sample with more than 94% hematite and granular quartz as the primary impurities. Rulyov et al. [54] revealed that the fraction containing only fine particles (0–5 mm) consumes approximately three times more flocculant than the fraction comprising exclusively coarse particles (5–10 mm) to achieve optimal dosages in terms of the relative mean size of flocs in quartz suspensions at various concentrations. For a fraction containing small and coarse particles (0–10 mm), the optimal dosage of flocculant is approximately the same as that for the fine fraction and weakly depends on the solids concentration.

### 3.2. Initial Settling Rates

Sedimentation experiments were conducted to determine the clarification performance of pulps, in which the settling rate is a critical parameter to define the unit area of thickeners. The factors that determine the settling rate of flocculated pulps were attributed to the size and fractal dimension of the aggregates generated by flocculation, the size of the primary particles, the initial concentration of pulp solids, the densities of the solid and liquid phases that make up the pulp, and liquid phase viscosity [35].

Figure 4 displays the results of the initial settling rate experiments of flocculated synthetic tailings at various flocculation times and kaolin contents for the following conditions: (A) fine quartz in process water, (B) fine quartz in seawater, (C) coarse quartz in process water, and (D) coarse quartz in seawater. The following findings were obtained:The sedimentation rate decreased with the increase in flocculation time for all coarse quartz-based tailings conditions (Figure 4C,D). However, for fine quartz-based tailings, the trend is only clear for tailings of Qz:Kao = 9:1. For tailings with higher kaolin contents, the sedimentation rate tends to be constant with flocculation time. The flocculation kinetics (Figure 3) revealed that the maximum size of the flocs was reached in the first 10–20 s. Subsequently, the destruction of the flocs was superimposed on the aggregation rate, resulting in smaller flocs as the mixing time was prolonged; consequently, smaller flocs produced a lower settling rate.The settling rate was clearly affected by the increase in the kaolin content in the tailings. Flocculation kinetics indicated a larger size of the flocs for the tailings with a higher kaolin content, which indicated an open and irregular structure because of the laminated morphology of the clay that, when associated, generates larger but mostly porous aggregates that ultimately prejudice the settling rate.The settling rate was slightly faster for tailings with process water than with seawater; the flocculation kinetics revealed that the size of the flocs was slightly larger in process water than in seawater, resulting in a higher sedimentation velocity for tailings with the first type of water.The settling rate was higher for tailings with coarse quartz than with fine quartz; with the same explanation as in the previous case, the flocculation kinetics revealed a larger floc size for coarse quartz.

### 3.3. Turbidity

Figure 5 displays the results of turbidity measurements of the supernatant of synthetic tailings flocculated at various flocculation times and kaolin contents for the following conditions: (A) fine quartz in process water, (B) fine quartz in seawater, (C) coarse quartz in process water, and (D) coarse quartz in seawater.

The highest levels of turbidity occur in tailings with lower kaolin content, coarse quartz, and process water. These results can be categorized into two categories: (1) the influence of the solid phase (kaolin content and quartz size) and (2) the influence of the liquid phase (type of water) of the tailings.

In the first case, turbidity is related to the settling rate. As in the previous section, the lower the kaolin content, the larger the size of the quartz, the higher the settling rate, and the higher the turbidity of the supernatant. This phenomenon can be explained by drag force interference, which acts in the opposite direction to that of sedimentation, where the magnitude of this force is proportional to the square of the settling rate. This phenomenon involves subjecting the flocs to a greater drag force, increasing floc erosion, which finally achieves solid-phase detachment that remains suspended in the supernatant.

In the second case, the quality of the tailings water influenced the supernatant turbidity, in which the seawater presented a reduced turbidity compared with the process water. The high ionic strength of seawater reduces the electrical double layer, producing an agglomeration of particles, especially fine ones, and increasing polymer adsorption. Nasser and James [55] performed a similar study for the flocculation of kaolin suspensions using polyacrylamides of various charge densities and molecular weights. The results revealed that the high negative zeta potentials of the particles treated with anionic polymers could be attributed to lower supernatant clarity than those treated with cationic polymers.

### 3.4. Fractal Dimension

The fractal dimension describes the morphological structure of the aggregates generated by the arrangements of the particles during the aggregation process; therefore, the nature of the particles characterized by their shape, size, and surface charge can be a determining factor of arrangements. The interaction force between the particles defines the modes of association and consequently aggregate structure, within which polymeric forces [37,41,56], hydrodynamic forces [36,37,57], and chemical conditions of the suspension [23,24] can be highlighted. Mathematically, the mass of an aggregate (m) is proportional to the characteristic length (L) of the same aggregate, raised to a power ($D_{f}$) that quantitatively describes the porosity and shape of the aggregate, known as the fractal length dimension of mass. This relationship can be expressed as follows:(3)$$m\propto {\left(L\right)}^{D_{f}}$$

The fractal dimension takes values between 1 and 3, where the first value represents an aggregate without spatial dimension, that is, it is equivalent to a point in space, whereas the second value represents an aggregate with a perfectly spherical and compact shape. By contrast, it is assumed that the number of primary particles within an aggregate (N) is related to the mass fractal dimension of the aggregate [58], according to the following:(4)$$N∼{\left(\frac{d}{d_{p}}\right)}^{D_{f}}$$
where $d_{p}$ is the size of the primary particles and d is the diameter (size) of the floc.

Figure 6 displays the fractal dimensions of synthetic tailing aggregates flocculated at various flocculation times and kaolin contents for the following conditions: (A) fine quartz in process water, (B) fine quartz in seawater, (C) coarse quartz in process water, and (D) coarse quartz in seawater.

The results revealed a clear trend, lower kaolin content, and lower fractal dimension for the four cases, whereas the other conditions did not significantly affect this parameter. The size of the flocs determined by the mean squared-weighted chord length delivered by the FBRM technology indicated higher values for tailings with higher kaolin content; however, sedimentation experiments indicated a lower settling rate for these tailings, which could be attributed to an open and irregular structure in shape (lower fractal dimension), a product of the “disorder” of aggregation between particles that implies the presence of kaolin because of its laminar and anisotropic structure.

This result is consistent with Vahedi and Gorczyca [17], who determined that aggregates can have the same size but different structures resulting from various particle arrangements. By contrast, the results indicated that the fractal dimension decreased with the flocculation time, which is consistent with previous studies [36,37], as a product of irreversible breakage caused by shear forces, which indicates subjecting the aggregates formed by the flocculation of the polymer bridge to prolonged mixing, concluding that the aggregates become progressively more irregular in shape and porous in structure. Additionally, the fractal dimension of the flocculated aggregates of the tailings composed of coarse quartz was lower than that of the tailings composed of fine quartz. This result was consistent with that of Leiva et al. [36,37], as finer particles are more likely to occupy empty spaces within the flocs, achieving more compact structures.

The reduction of the fractal dimension with an increase in the kaolin content was less intense for seawater than for process water for both quartz sizes. For fine quartz and at 20 s of flocculation, the value of the fractal dimension changed from $D_{f}$ = 2.27 (Qz:Kao = 9:1) to $D_{f}$ = 2.03 (Qz:Kao = 7:3) in process water and from $D_{f}$ = 2.26 (Qz:Kao = 9:1) to $D_{f}$ = 2.04 (Qz:Kao = 7:3) in seawater. For coarse quartz and at 20 s of flocculation, the value of the fractal dimension changed from $D_{f}$ = 2.26 (Qz:Kao = 9:1) to $D_{f}$ = 2.06 (Qz:Kao = 7:3) in process water and from $D_{f}$ = 2.18 (Qz:Kao = 9:1) to $D_{f}$ = 2.07 (Qz:Kao = 7:3) in seawater. This phenomenon could be attributed to the greater coagulant power of seawater, which agglomerates fine particles before flocculant addition.

## 4. Conclusions

The characterization of the aggregates generated by the flocculation of synthetic tailings composed of two mineralogical species, quartz and kaolin, in various liquid media was analyzed in terms of the variation of the quartz—kaolin ratio and flocculation duration.

Flocculation kinetics revealed that tailings with a higher presence of kaolin achieved larger aggregate sizes because of the particle arrangement that represents the laminar morphology of kaolin. The results indicated that the maximum size of the aggregates reached between 10 and 20 s of flocculation for all evaluated physicochemical conditions. However, the presence of kaolin negatively affected the sedimentation rate, decreasing its value with the increase in kaolin concentration.

From the experimental results, the model suggested that the fractal dimension decreased with an increase in kaolin, indicating the formation of structural aggregates that were mostly irregular in shape and open, which translates into greater porosity. This characteristic of the highly clayey flocs resulted in a sediment with a lower concentration of solids and a higher water content.

Other factors evaluated, such as flocculation time, water quality, and quartz size, did not significantly influence flocculation performance compared with kaolin presence. However, seawater produced a clearer supernatant than process water because of its higher ionic strength, leading to prior coagulation of the particles.

## Figures and Tables

**Figure 1 polymers-16-01441-f001:**
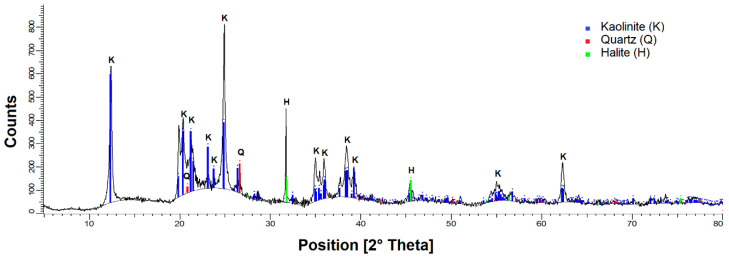
X-ray diffraction spectrum of kaolin particles.

**Figure 2 polymers-16-01441-f002:**
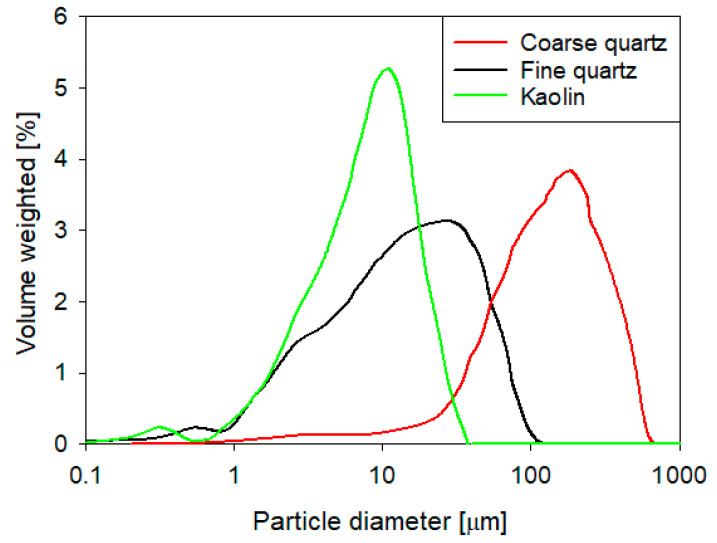
Volume-weighted size distribution of quartz and kaolin particles.

**Figure 3 polymers-16-01441-f003:**
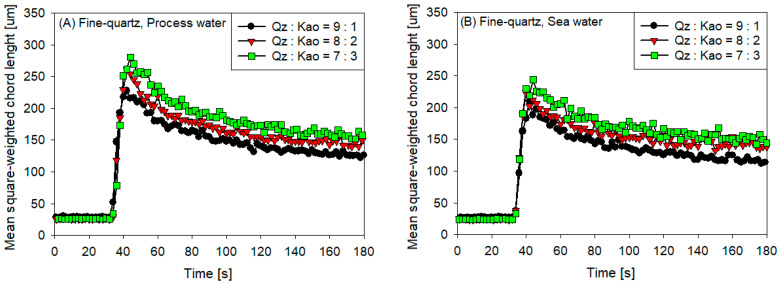
Flocculation kinetics of synthetic tailings flocculated at 25 g/t of SNF604, pH 7.5, stirring at 180 rpm, and solids concentration of 10% weight for various kaolin contents: (**A**) fine quartz in process water, (**B**) fine quartz in seawater, (**C**) coarse quartz in process water, and (**D**) coarse quartz in seawater. The flocculant solution is added 30 s after starting the measurement.

**Figure 4 polymers-16-01441-f004:**
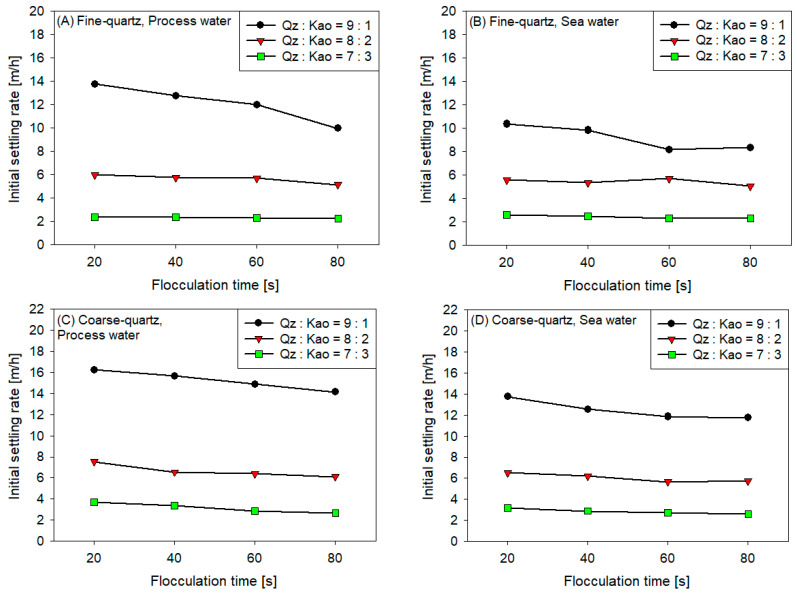
Initial settling rates of synthetic tailings flocculated at 25 g/t of SNF604, pH 7.5, stirring at 180 rpm, and solids concentration of 10% weight for various kaolin contents: (**A**) fine quartz in process water, (**B**) fine quartz in seawater, (**C**) coarse quartz in process water, and (**D**) coarse quartz in seawater.

**Figure 5 polymers-16-01441-f005:**
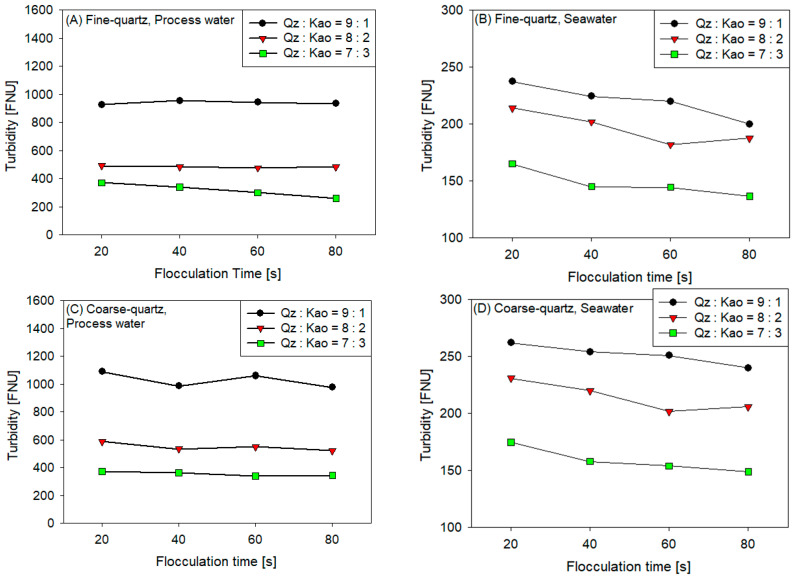
Turbidity of the supernatant of synthetic tailings flocculated at 25 g/t of SNF604, pH 7.5, stirring at 180 rpm, and solids concentration of 10% weight for various kaolin contents: (**A**) fine quartz in process water, (**B**) fine quartz in seawater, (**C**) coarse quartz in process water, and (**D**) coarse quartz in seawater.

**Figure 6 polymers-16-01441-f006:**
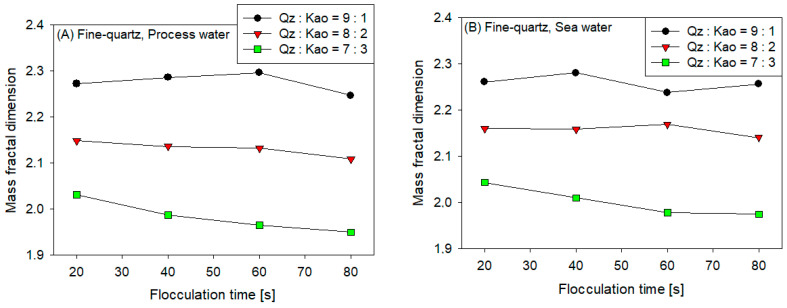
Mass fractal dimension of synthetic tailing aggregates flocculated at 25 g/t SNF604, pH 7.5, 180 rpm stirring, and 10 wt% solids concentration for various flocculation times and kaolin contents: (**A**) fine quartz in process water, (**B**) fine quartz in seawater, (**C**) coarse quartz in process water, and (**D**) coarse quartz in seawater.

**Table 1 polymers-16-01441-t001:** Composition of artificial seawater according to Mobin and Shabnam, 2011.

Component	Concentration (g/L)
NaCl	24.53
MgCl_2_ × 6H_2_O	11.10
Na_2_SO_4_	4.09
CaCl_2_	1.16
KCl	0.69
NaHCO_3_	0.20
KBr	0.10
H_3_BO_3_	0.03

**Table 2 polymers-16-01441-t002:** Known and constant parameters for all working pulps.

Parameter	Magnitude
g $\left(ms^{-2}\right)$	9.81
$\mu_{IW}$ $\left(Nsm^{-2}\right)$	0.001021
$\mu_{SW}$ $\left(Nsm^{-2}\right)$	0.001077
$\emptyset_{s}$	0.0413

## Data Availability

Data are contained within the article.

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
