# Peer review of "Clay Tailings Flocculated in Seawater and Industrial Water: Analysis of Aggregates, Sedimentation, and Supernatant Quality"

_polymers, 2024, doi:10.3390/polym16101441_

Round 1
Reviewer 1 Report
Comments and Suggestions for Authors
Comments
In this work, the authors reported the effect of the kaolinite concentration on the restructuring of aggregates flocculated with an anionic polyacrylamide, which is of great practical importance in mining wastewater treatment and resource recovery. In my view, the following issues need to be addressed before consideration for publication.
1. Please provide a further explanation of the phenomenon in Fig. 3d.
2. Consideration can be given to adding some 3D graphics or animation to visualise the dynamics of particle aggregation during flocculation.
3. Please explore the effect of different types and sources of kaolin on flocculation.
4. Please explore the effect on flocculation at different pH values.
Comments on the Quality of English Language
Minor editing of English language required
Author Response
Response Reviewer 1
Comments and Suggestions for Authors
Comments
In this work, the authors reported the effect of the kaolinite concentration on the restructuring of aggregates flocculated with an anionic polyacrylamide, which is of great practical importance in mining wastewater treatment and resource recovery. In my view, the following issues need to be addressed before consideration for publication.
- Please provide a further explanation of the phenomenon in Fig. 3d.
To answer this suggestion, the next text was added on line 336
However, Figure 4(D) shows that the floc size for tailings Qz:Kao = 7:3 is smaller than the flocs for tailings Qz:Kao = 8:2 after reaching the optimal size for both cases. This phenomenon, which differs from the other behaviors, can be explained by two reasons: (i) larger quartz produces less probability of adsorption on kaolin particles, generating weaker quartz-kaolin aggregates that are easier to break as a result of the hydrodynamic forces generated by the mixing intensity, (ii) seawater favors cationic bridges between particles, which in the case of kaolin-kaolin particles implies a structure in the form of a house of cards that, when agglomerated by polymeric bridges, generates more porous structures that are easier to break.
- Consideration can be given to adding some 3D graphics or animation to visualise the dynamics of particle aggregation during flocculation.
- Please explore the effect of different types and sources of kaolin on flocculation.
Done
- Please explore the effect on flocculation at different pH values.
We consider that these physicochemical conditions can be studied in future works. There is very little literature that analyzes/evaluates these effects on tailings flocculation.

Reviewer 2 Report
Comments and Suggestions for Authors
This study deals with the flocculation of kaolin under various conditions. Kaolin is a prominent clay and well suited for model investigations with clay. The article is fluently written, the experiments were performed properly and the results are presented in a well structured way. Nonetheless, some revision is required as indicated below; in particular some statements do not hold.
Figure 2 is mentioned on line 124 but Figure 1 only on line 127, i.e. Figure 1 is mentioned after Figure 2 in the text. Therefore those Figures should be re-numbered accordingly.
Line 127- 128: It is mentioned that the samples contains a majority of kaolinite above 10%. However, a fraction of 10% is not a majority. In fact it appears to me from Figure 1 that the fraction of kaolinite is much higher than 10%, so I propose to check if my impression could be right and if yes to provide an appropriate higher estimation, or otherwise to remove the statement that the sample contains a majority of kaolinite.
Line 234 – 235: “… quartz has a crystal structure composed exclusively of [𝑆𝑖𝑂4]4− tetrahedra”: This description of quartz is confusing and in fact impossible since the negative charges of the tetrahedra are not balanced by positively charged counterions. The description of the crystal structure should be changed accordingly.
Line 235: “all 𝑂s united in a three-dimensional network”: The expression “Os” is sloppy and should be replaced by “O atoms”.
I was not able to find Figure 3 in the text. It is not clear to me why this figure is shown.
Line 291 – 292: “The settling rate decreased with the increase in the flocculation time for all conditions”: This is not the case in Figure 4 (A) in the case of Qz : Kao = 7:3, in Figure 4 (B) in the case of Qz : Kao = 7:3 and Qz : Kao = 8:3 and hardly evident in some other cases.
Line 292 – 293: “the maximum size of the flocs was reached in the first 15–20 s”: This is not evident from Figure 4 at all since values between 15-20 s are missing in all plots.
Line 317 – 318: “The highest levels of turbidity occur for tailings with lower kaolin content “: This effect is not significanr in sea water. Here the changes between the different Qz : Kao ratios are not relevant.
Line 365 – 366: “The results revealed a clear trend, lower kaolin content, and lower fractal dimension for the four cases”: This trend is not relevant in Figure 6 (D) and only partially evident in Figure 6 (C).
Author Response
Response Reviewer 2
Comments and Suggestions for Authors
This study deals with the flocculation of kaolin under various conditions. Kaolin is a prominent clay and well suited for model investigations with clay. The article is fluently written, the experiments were performed properly and the results are presented in a well structured way. Nonetheless, some revision is required as indicated below; in particular some statements do not hold.
- Figure 2 is mentioned on line 124 but Figure 1 only on line 127, i.e. Figure 1 is mentioned after Figure 2 in the text. Therefore those Figures should be re-numbered accordingly.
The figures were rearranged in the order mentioned.
- Line 127- 128: It is mentioned that the samples contains a majority of kaolinite above 10%. However, a fraction of 10% is not a majority. In fact it appears to me from Figure 1 that the fraction of kaolinite is much higher than 10%, so I propose to check if my impression could be right and if yes to provide an appropriate higher estimation, or otherwise to remove the statement that the sample contains a majority of kaolinite.
Figure 1 (now Figure 2) corresponds to a diffractogram that indicates the presence of species and is associated with the mentioned software which does not indicate the exact composition of each mineral, it only indicates whether an individual species is above or below 10%, therefore, it is not possible to provide a higher estimate. It is also not possible to remove the statement that the sample contains a majority of kaolinite since that is the information provided by the software. In summary, the software indicates that kaolinite is above 10% composition and is mostly present, it does not provide more than this.
- Line 234 – 235: “… quartz has a crystal structure composed exclusively of [???4]4−tetrahedra”: This description of quartz is confusing and in fact impossible since the negative charges of the tetrahedra are not balanced by positively charged counterions. The description of the crystal structure should be changed accordingly.
The mentioned description was complemented with the following expression to close the idea:
“Thus, the formula is SiO2 and the atoms are arranged in a trigonal symmetry.”
- Line 235: “all ?s united in a three-dimensional network”: The expression “Os” is sloppy and should be replaced by “O atoms”.
Done. “?s” was replaced by “O atoms”.
- I was not able to find Figure 3 in the text. It is not clear to me why this figure is shown.
Yes, it is. It was on line 276 of the previous manuscript and is now on line XXX of the revised version of the manuscript. Figure 3 corresponds to “flocculation kinetics” which basically shows the variation in the size of the flocs with time where the addition of flocculant occurs 30 s after the measurement began. Regarding the latter, the following text is added in the figure legend: “The flocculant solution is added 30 s after starting the measurement”, even though it is already mentioned in section 2.2, which in the previous version said “Materials” but which actually corresponds to “Flocculation kinetics”
- Line 291 – 292: “The settling rate decreased with the increase in the flocculation time for all conditions”: This is not the case in Figure 4 (A) in the case of Qz : Kao = 7:3, in Figure 4 (B) in the case of Qz : Kao = 7:3 and Qz : Kao = 8:3 and hardly evident in some other cases.
The comment basically refers to the trend, that is, for all conditions the settling rate goes down with flocculation time. Now, the points that you mention correspond to “outer data” only (points that escape the trend which are exceptions). In the case of pulps with higher kaolin content the trend is not very evident, but it happens.
- Line 292 – 293: “the maximum size of the flocs was reached in the first 15–20 s”: This is not evident from Figure 4 at all since values between 15-20 s are missing in all plots.
This comment refers to what was observed in Figure 3 (see response to comment 5), although there was a modification since the mentioned range was changed to 10-20 s. Figure 3 is also cited again.
- Line 317 – 318: “The highest levels of turbidity occur for tailings with lower kaolin content “: This effect is not significanr in sea water. Here the changes between the different Qz : Kao ratios are not relevant.
In our observation, the kaolin content is relevant in seawater. If we take the average turbidity of the ratios Qz:Kao = 9:1, Qz:Kao = 8:2 and Qz:Kao = 7:3 for fine quartz, values of around 220, 196 and 148 FNU are obtained, respectively. This implies that if we go from the ratio 9:1 to 8:2 and from 9:1 to 7:3 the turbidity decreases by 11% and 33%, respectively. Although these decreases are less pronounced than in process water, it is still a considerable decrease when it comes to improving the quality of the recovered water.
- Line 365 – 366: “The results revealed a clear trend, lower kaolin content, and lower fractal dimension for the four cases”: This trend is not relevant in Figure 6 (D) and only partially evident in Figure 6 (C).
Although in the cases you mention they are less relevant than in the case of the other figures, the trend is also true.

Reviewer 3 Report
Comments and Suggestions for Authors
The article needs minor revisions. Comments are marked in the attached file.

Author Response
- Line 40-41: It is advisable to provide literature references.
Four new references have been added.
- Deniz, V. Dewatering of barite clay wastewater by inorganic coagulants and co-polymer flocculants. Physicochem Probl Miner Process 2015, 51, 351-364, doi:10.5277/ppmp150131.
- Blanco, A.; Fuente, E.; Negro, C.; Tijero, J. Flocculation Monitoring: Focused Beam Reflectance Measurement as a Measurement Tool. Can J Chem Eng 2002, 80, 1-7, doi:10.1002/cjce.5450800403.
- Mpofu, P.; Addai-Mensah, J.; Ralston, J. Influence of hydrolyzable metal ions on the interfacial chemistry, particle interactions, and dewatering behavior of kaolinite dispersions. J Colloid Interface Sci 2003, 261, 349-359, doi:10.1016/S0021-9797(03)00113-9.
- Penner, D.; Lagaly, G. Influence of Organic and Inorganic Salts on the Coagulation of Montmorillonite Dispersions. Clays Clay Miner 2000, 48, 246-255, doi:10.1346/CCMN.2000.0480211.
- Line 124: Figures should be numbered in the order they are mentioned in the text.
The figures were rearranged in the order mentioned in the text.
- Line 127: see comment in line 124
See comment 2
- Line 138: not only kaolin
Yes, it is. Kaolin is made up of those three minerals mentioned in the graph.
- Page 4: Specify the parameter ranges used in the performed experiments.
There are no ranges for the masses used, they are exact quantities with very low variation. The scale used provides 4 decimal digits where the first two were zeros and the last two were different from zero, giving the low variation. For example, if 25 g of solids were weighed, the solids were added up to 25.00XX g where XX represent values different from zero, thus giving a quite low error. Only pH has a range due to its greater sensitivity.
- Line 293: You can't talk about such an range if the first measurement was taken after 20s.
In lines XXX-XXX of section 2.2 is mentioned that the flocculant is added after 30 s once the measurement has started. The range mentioned here corresponds to the range that the maximum size of the flocs is reached after the flocculant has been added (i.e., when the flocculation process begins), not with respect to the start of the measurement.
On the other hand, the range was modified to 10-20 s.
- Line 399-407: This information should not be in the paragraph "conclusions"
From line 402 to 407 was removed, however from 309 to 401 was kept because it is introductory to the conclusion.
- Line 411: see comment in line 293.
See comment 6.

Round 2
Reviewer 2 Report
Comments and Suggestions for Authors
The answers number 1, 2, 4, 5 and 7 are okay, however, it appears to me that answers 3, 6, 8 and 9 are not satisfactory:
Answer 3: When the structure of quartz is “composed exclusively of [𝑆𝑖𝑂4]4− tetrahedra the resulting formula is not SiO2 but a negatively charged substance of the type [SiO2]z-. Accordingly, the principle of electroneutality is violated. Negative charges in an isolated chemical compound always have to compensated by cations, otherwise the compound does not exist at ambient conditions.
Answer 6: I cannot follow the authors’ reply. In Figure 4 (A) and 4 (B) only in the case of Qz : Kao = 9 : 1 there is a clear trend in decrease of the settling rate with flocculation time. In the other cases the flocculation time is essentially constant within the experimental limits.
Answer 8: The author claim that the decreases in sea water are considerable. However, it is clear that the precision of the determination of the turbidity has its limitations, and the curves are so close that I do not consider the differences significant. Notably, in Figure 5 (B) 3 out of 4 points of the Qz : Kao = 9 : 1 and 8 : 2 are almost identical. If the authors calin that the differences are significant then this ha to be shown clearly by 95% confidence intervals of the data points. If this is not shown then the claim of significant differences is not reliable.
Answer 9: Again, the claim that the trends in Figure 6 (C) and 6 (D) are significant have to be substantiated again with 95% confidence intervals as in particular two curves in Figure 6 (D) almost overlap, and the differences between the curves in Figure 6 (C) and Figure 6 (D) are not pronounced in general.
Author Response
Comments and Suggestions for Authors
The answers number 1, 2, 4, 5 and 7 are okay, however, it appears to me that answers 3, 6, 8 and 9 are not satisfactory:
(2° Round): Answer 3: When the structure of quartz is “composed exclusively of [???4]4− tetrahedra the resulting formula is not SiO2 but a negatively charged substance of the type [SiO2]z-. Accordingly, the principle of electroneutality is violated. Negative charges in an isolated chemical compound always have to compensated by cations, otherwise the compound does not exist at ambient conditions.
We appreciate your valuable comments, which have allowed us to clarify the description of the structure and electrical neutrality of quartz in our manuscript. We have revised and modified our previous description as the mention of the [???4]4− tetrahedra was confusing. Quartz, with the chemical formula SiO2, has a crystalline structure where each silicon atom is surrounded by four oxygen atoms, forming a SiO4 tetrahedron. It is crucial to highlight that these tetrahedra are interconnected so that each oxygen atom is shared between two tetrahedra, effectively neutralizing the charge without additional cations. This arrangement results in a global unitary formula of SiO2, reflecting a covalently bonded network that is electrically neutral.
We have included the following in the manuscript:
"In the crystalline structure of quartz, each silicon atom is tetrahedrally coordinated by four oxygen atoms. These [SiO4] tetrahedra do not carry a net negative charge within the complete context of the quartz structure, as the oxygen atoms are shared between adjacent tetrahedra, resulting in a globally neutral chemical composition of SiO2."
(1° Round): 3. Line 234 – 235: “… quartz has a crystal structure composed exclusively of [???4]4− tetrahedra”: This description of quartz is confusing and in fact impossible since the negative charges of the tetrahedra are not balanced by positively charged counterions. The description of the crystal structure should be changed accordingly.
The mentioned description was complemented with the following expression to close the idea:
“Thus, the formula is SiO2 and the atoms are arranged in a trigonal symmetry.”
(2° Round): Answer 6: I cannot follow the authors’ reply. In Figure 4 (A) and 4 (B) only in the case of Qz : Kao = 9 : 1 there is a clear trend in decrease of the settling rate with flocculation time. In the other cases the flocculation time is essentially constant within the experimental limits.
The comment was complemented with the following expression:
The sedimentation rate decreased with the increase in the flocculation time for all coarse quartz-based tailings conditions (Figure 4(C) and Figure 4(D)). However, for fine quartz-based tailings, the trend is only clear for tailings of Qz:Kao = 9:1. For tailings with higher kaolin contents, the sedimentation rate tends to be constant with flocculation time.
(1° Round): 6. Line 291 – 292: “The settling rate decreased with the increase in the flocculation time for all conditions”: This is not the case in Figure 4 (A) in the case of Qz : Kao = 7:3, in Figure 4 (B) in the case of Qz : Kao = 7:3 and Qz : Kao = 8:3 and hardly evident in some other cases.
The comment basically refers to the trend, that is, for all conditions the settling rate goes down with flocculation time. Now, the points that you mention correspond to “outer data” only (points that escape the trend which are exceptions). In the case of pulps with higher kaolin content the trend is not very evident, but it happens.
(2° Round): Answer 8: The author claim that the decreases in sea water are considerable. However, it is clear that the precision of the determination of the turbidity has its limitations, and the curves are so close that I do not consider the differences significant. Notably, in Figure 5 (B) 3 out of 4 points of the Qz : Kao = 9 : 1 and 8 : 2 are almost identical. If the authors calin that the differences are significant then this ha to be shown clearly by 95% confidence intervals of the data points. If this is not shown then the claim of significant differences is not reliable.
We have decided to reduce the turbidity scale for Figures 5(B) and 5(D) (from 100 to 300 FNU) to highlight the difference between kaolin concentrations.
(1° Round): 8. Line 317 – 318: “The highest levels of turbidity occur for tailings with lower kaolin content “: This effect is not significanr in sea water. Here the changes between the different Qz : Kao ratios are not relevant.
In our observation, the kaolin content is relevant in seawater. If we take the average turbidity of the ratios Qz:Kao = 9:1, Qz:Kao = 8:2 and Qz:Kao = 7:3 for fine quartz, values of around 220, 196 and 148 FNU are obtained, respectively. This implies that if we go from the ratio 9:1 to 8:2 and from 9:1 to 7:3 the turbidity decreases by 11% and 33%, respectively. Although these decreases are less pronounced than in process water, it is still a considerable decrease when it comes to improving the quality of the recovered water.
(2° Round): Answer 9: Again, the claim that the trends in Figure 6 (C) and 6 (D) are significant have to be substantiated again with 95% confidence intervals as in particular two curves in Figure 6 (D) almost overlap, and the differences between the curves in Figure 6 (C) and Figure 6 (D) are not pronounced in general.
Here we have also decided to reduce the mass fractal dimension scale for Figures 6(A-D) (from 1.9 to 2.4) to highlight the difference between the kaolin concentrations.
(1° Round): 9. Line 365 – 366: “The results revealed a clear trend, lower kaolin content, and lower fractal dimension for the four cases”: This trend is not relevant in Figure 6 (D) and only partially evident in Figure 6 (C).
Although in the cases you mention they are less relevant than in the case of the other figures, the trend is also true.

Round 3
Reviewer 2 Report
Comments and Suggestions for Authors
The structure of quartz is now correctly described, and the modification of the graphs show now the trends better except in Figure 3 (C) where the curves of Qz : Kao = 9 : 1 and 8 : 2 clearly overlap. Thus, there is no trend from Qz : Kao = 9 : 1 to 7 : 3.
Author Response
Comments and Suggestions for Authors:
The structure of quartz is now correctly described, and the modification of the graphs show now the trends better except in Figure 3 (C) where the curves of Qz : Kao = 9 : 1 and 8 : 2 clearly overlap. Thus, there is no trend from Qz : Kao = 9 : 1 to 7 : 3.
We appreciate your comment. To solve this appreciation issue, a zoom image was created and placed in Figure 3(C) to show that there is no overlap in any of the three curves, especially in the maximum floc size of each curve. Therefore, Figure 3(C) was modified.
